∂ | **Open Peer Review** | Antimicrobial Chemotherapy | *Research Article*

# Rapid emergence of resistance to broad-spectrum direct antimicrobial activity of avibactam

Michelle Nägeli,[1] Shade Rodriguez,[1] Aimee Iradukunda,[1] Abigail L. Manson,[2] Ashlee M. Earl,[2] Thea Brennan-Krohn[1,3,4]

**ABSTRACT** Avibactam (AVI) is a diazabicyclooctane (DBO) β-lactamase inhibitor used clinically in combination with ceftazidime. At concentrations higher than those typically achieved *in vivo*, it also has broad-spectrum direct antibacterial activity against *Enterobacterales* strains, including metallo-β-lactamase-producing isolates, mediated by inhibition of penicillin-binding protein 2 (PBP2). This activity has some mechanistic similarities to that of more potent novel DBOs (zidebactam and nacubactam) in late clinical development. We found that resistance to AVI emerged readily, with a mutation frequency of $2 \times 10^{-6}$ to $8 \times 10^{-5}$. Whole-genome sequencing of resistant isolates revealed a heterogeneous mutational target that permitted bacterial survival and replication despite PBP2 inhibition, in line with prior studies of PBP2-targeting drugs. While such mutations are believed to act by upregulating the bacterial stringent response, we found a similarly high mutation frequency in bacteria deficient in components of the stringent response, although we observed a different set of mutations in these strains. Although avibactam-resistant strains had increased lag time, suggesting a fitness cost that might render them less problematic in clinical infections, there was no statistically significant difference in growth rates between susceptible and resistant strains. The finding of rapid emergence of resistance to avibactam as the result of a large and complex mutational target adds to our understanding of resistance to PBP2-targeting drugs and has potential implications for novel DBOs with potent direct antibacterial activity, which are being developed with the goal of expanding cell wall-active treatment options for multidrug-resistant gram-negative infections.

**IMPORTANCE** Avibactam (AVI) is the first in a class of novel β-lactamase inhibitor antibiotics called diazabicyclooctanes (DBOs). In addition to its ability to inhibit bacterial β-lactamase enzymes that can destroy β-lactam antibiotics, we found that AVI had direct antibacterial activity, at concentrations higher than those used clinically, against even highly multidrug-resistant bacteria. This activity is the result of inhibition of the bacterial enzyme penicillin-binding protein 2 (PBP2). Resistance to other drugs that inhibit PBP2 occurs through mutations that involve upregulation of the bacterial "stringent response" to stress. We found that bacteria developed resistance to AVI at a high rate, as a result of mutations in stringent response genes. We also found that bacteria with impairments in the stringent response could still develop resistance to AVI through different mutations. Our findings indicate the importance of studying how resistance will emerge to newer, more potent DBOs in development and early clinical use.

**KEYWORDS** avibactam, antimicrobial resistance, antibiotic resistance, penicillin-binding proteins, diazabicyclooctane, gram-negative bacteria

β-Lactamase inhibitors have a long history in the arms race between humans and microbes. By defending β-lactam antibiotics against bacterial β-lactamase enzymes, they extend the activity spectrum of β-lactams, which are valued as first-line

Address correspondence to Thea Brennan-Krohn, tkrohn@bidmc.harvard.edu.

Michelle Nägeli and Shade Rodriguez contributed equally to this article. Author order was determined based on seniority.

The authors declare no conflict of interest.

See the funding table on p. 20.

therapeutic agents because of their safety, efficacy, and long clinical track record (1). β-Lactamase inhibitors have grown increasingly important in the era of multi-drug resistance, as broad-spectrum β-lactamases, including carbapenemases, have emerged as a key β-lactam resistance mechanism among gram-negative bacteria (2). For decades, all β-lactamase inhibitors in clinical use were themselves β-lactam compounds that serve as "suicide inhibitors" of β-lactamase enzymes (3). Despite their β-lactam structure, these compounds have minimal intrinsic antibacterial activity (with a few exceptions, most notably sulbactam, which is active against *Acinetobacter baumannii* [3]). In 2015, the first non-β-lactam β-lactamase inhibitor, avibactam (AVI), was approved by the United States Food and Drug Administration (FDA) as a combination product with ceftazidime. Avibactam was the first β-lactamase inhibitor with activity against serine carbapenemase enzymes, including *Klebsiella pneumoniae* carbapenemases (KPCs), thus rendering ceftazidime-avibactam the first β-lactam-based agent that could be used to treat infections caused by carbapenem-resistant *Enterobacterales*.

Avibactam is a diazabicyclooctane (DBO) compound (Fig. 1). It exerts its β-lactamase inhibitor activity through covalent, reversible binding to serine β-lactamases such as KPCs (4), although it does not inhibit metallo-β-lactamase carbapenemases (MBLs) (5). We observed that AVI also has direct *in vitro* antimicrobial activity against *Enterobacterales* isolates, including MBL-producing strains, a phenomenon mediated by its binding of penicillin-binding protein 2 (PBP2) (6), one of the classes of transpeptidase enzymes involved in bacterial peptidoglycan synthesis that constitute the targets of β-lactam drugs (7). Despite prior reports in the literature of direct AVI activity, the drug is typically described as lacking intrinsic antibacterial activity (8, 9), probably because serum levels achieved with standard doses of ceftazidime-avibactam are unlikely to be high enough to exert significant direct activity (10) given the range of AVI MICs (8–32 µg/mL). In recent years, however, DBOs with much more potent direct antimicrobial activity have been developed. One such compound, zidebactam, is currently undergoing phase 3 trials in combination with cefepime (11, 12). The spectrum of zidebactam's direct activity

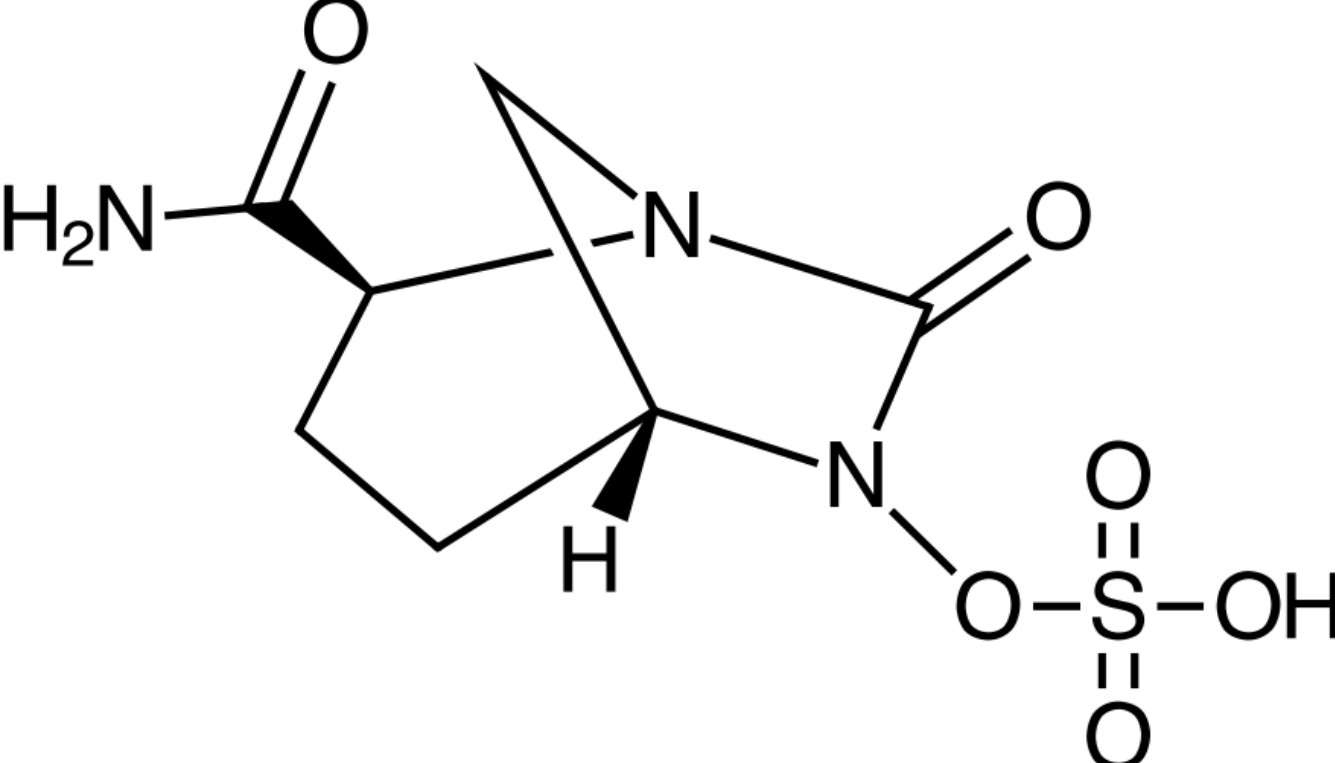

**FIG 1** Chemical structure of avibactam.

includes *Pseudomonas aeruginosa*, and case reports have described the compassionate use of cefepime-zidebactam for the treatment of MBL-producing *Pseudomonas aeruginosa* infections (13, 14). Because zidebactam, like avibactam, does not inhibit MBL enzymes, the activity of cefepime-zidebactam against MBL-producing strains results primarily from the intrinsic PBP2-inhibiting activity of zidebactam (a β-lactam "enhancer" effect has also been described, in which strains resistant to both components of a DBO-β-lactam pair are still susceptible to the combination; this activity is thought to reflect a synergy between residual PBP inhibitory activity of the two drugs [11, 15]) We observed a high rate of resistance mutation frequency to AVI, a phenomenon that is familiar from experience with the β-lactam mecillinam (amdinocillin), the only clinically available drug that exclusively targets PBP2 (16), and has also been observed in early investigations of nacubactam, another DBO with potent direct antibacterial activity (17). Another DBO with potent direct activity against *Enterobacterales*, durlobactam, is clinically available in combination with sulbactam; in this combination, approved for use against *Acinetobacter*, durlobactam functions primarily as a β-lactamase inhibitor (18).

In this paper, we describe the rapid emergence of stable, high-level resistance to direct AVI activity, which appears to result from a high frequency of resistance-conferring mutations and causes cross-resistance to other PBP2-targeting drugs (because no clinical interpretive criteria for direct avibactam activity exist, we use the term "AVI resistance" (AVI-R) throughout to refer to strains that developed AVI MICs of ≥8× the starting MIC upon exposure to AVI, while the parent strains are referred to as "AVI susceptible"). We also characterize these mutations through whole-genome sequencing. As is the case with mecillinam and nacubactam, AVI resistance involves a large and heterogeneous mutational target, with mutations in different AVI-resistant strains occurring in numerous different genes. The mechanism of resistance to mecillinam and nacubactam is believed to involve upregulation of the stringent response, resulting in compensatory changes that allow bacterial cells to tolerate PBP2 inhibition (16, 19), and our sequencing results, as well as morphological analysis of bacterial cells, demonstrate the same phenomenon with AVI. Interestingly, however, we observed a similarly high mutation frequency in strains deficient in components of the stringent response compared to their parent strains. Our results emphasize the importance of further study of resistance to novel DBO agents as they approach clinical availability in order to determine whether similar patterns of resistance may be observed with these compounds.

## RESULTS

### Avibactam has broad-spectrum activity against *Enterobacterales*

Avibactam MICs of 74 gram-negative bacterial isolates, enriched for carbapenemase-producing organisms, were tested using the digital dispensing method (DDM; Table 1; Table S1). MICs ranged from 4 to >64 µg/mL. $MIC_{50}$ and $MIC_{90}$ for *Enterobacterales* ($n = 54$) were 16 µg/mL and 64 µg/mL, respectively, while all *Pseudomonas aeruginosa* and *Acinetobacter baumannii* isolates ($n = 20$) had MICs of >64 µg/mL. Among isolates with

**TABLE 1** Avibactam MICs

| Category | $MIC_{50}$ | $MIC_{90}$ | MIC range |
| --- | --- | --- | --- |
| | | µg/mL | |
| All strains ($n = 74$) | 32 | 64 | 4 to >64 |
| *Enterobacterales* ($n = 54$) | 16 | 64 | 4 to >64 |
| E. coli ($n = 15$) | 8 | 64 | 4 to >64 |
| K. pneumoniae ($n = 25$) | 16 | 64 | 4 to >64 |
| Other *Enterobacterales* ($n = 14$) | 16 | 64 | 16 to >64 |
| Carbapenemase-producers (non-MBL; $n = 16$) | 8 | 64 | 4 to >64 |
| MBL-producers ($n = 33$) | 16 | 64 | 8 to >64 |
| *Pseudomonas aeruginosa* ($n = 5$) | >64 | >64 | >64 |
| *Acinetobacter baumannii* ($n = 15$) | >64 | >64 | >64 |

low AVI MICs (8 µg/mL) were two extremely multidrug-resistant strains: *K. pneumoniae* FDA-CDC 0636 (the pan-resistant "Nevada" strain, which encodes an NDM-1 metallo-β-lactamase [20, 21]) and *Escherichia coli* ARLG 2829 (the first strain identified in the US containing both a carbapenemase [NDM-5] and a mobile colistin resistance gene [*mcr-1*] [22]). To further investigate AVI activity against these isolates, time-kill studies were performed with AVI concentrations of 8, 16, 32, 64, and 128 µg/mL (Fig. 2). At the MIC for both strains (8 µg/mL), growth was inhibited through 6 hours, but there was no significant decrease in colony count. At higher concentrations of AVI, cell counts fell by 0.5–3.3 $\log_{10}$ CFU/mL from starting inoculum by 6 hours, but then began to increase. By 24 hours, cell density had increased by 2.0–3.0 $\log_{10}$ CFU/mL from starting inoculum even in cultures treated with AVI 128 µg/mL (16× MIC), although final colony counts were lower than in the untreated growth control.

## Avibactam activity is mediated by inhibition of PBP2

Although AVI is not a β-lactam compound, it is known to bind to and inhibit PBP2 (6, 23). Inhibition of different PBPs induces distinct morphological changes, with PBP2 inhibition in *Enterobacterales* resulting in the generation of enlarged, rounded cells (24). Serial Gram stain images of bacteria treated with AVI as in time-kill experiments were obtained under oil immersion magnification. At concentrations at and above the MIC, cells developed the distinctly rounded and enlarged appearance classically observed in bacteria treated with PBP2-inhibiting drugs (Fig. 3a). Notably, bacterial cells that have developed resistance to PBP2 targeting drugs such as mecillinam and nacubactam through non-β-lactamase-mediated mechanisms exhibit rounding during treatment with the drug, even though they are still able to survive and replicate (15, 25, 26), as resistance typically involves compensatory mutations in genes other than PBP2. To assess whether the same effect occurred with AVI, bacteria were grown with 128 µg/mL AVI as in time-kill experiments, subcultured overnight on antibiotic-free media, and then grown again for 24 hours in liquid culture containing AVI at 8 µg/mL and at 128 µg/mL. Although the AVI MIC, performed in parallel with the growth curve experiment, was confirmed as >256 µg/mL and the resistant cells grew to within 0.5 $\log_{10}$ CFU/mL of the untreated resistant strain by 24 hours (Fig. 4), Gram stain images at 3 and 24 hours showed rounded morphology of the cells at both AVI concentrations (Fig. 3b), indicating preservation of active PBP2 inhibition even in bacteria able to survive and replicate during AVI exposure.

Cross-resistance between AVI and other PBP-targeting drugs was assessed using two broadly β-lactam-susceptible isolates (*K. pneumoniae* BIDMC 22 and *E. coli* BIDMC 49A). Following growth for 24 hours in AVI 128 µg/mL, the MICs of mecillinam, a β-lactam

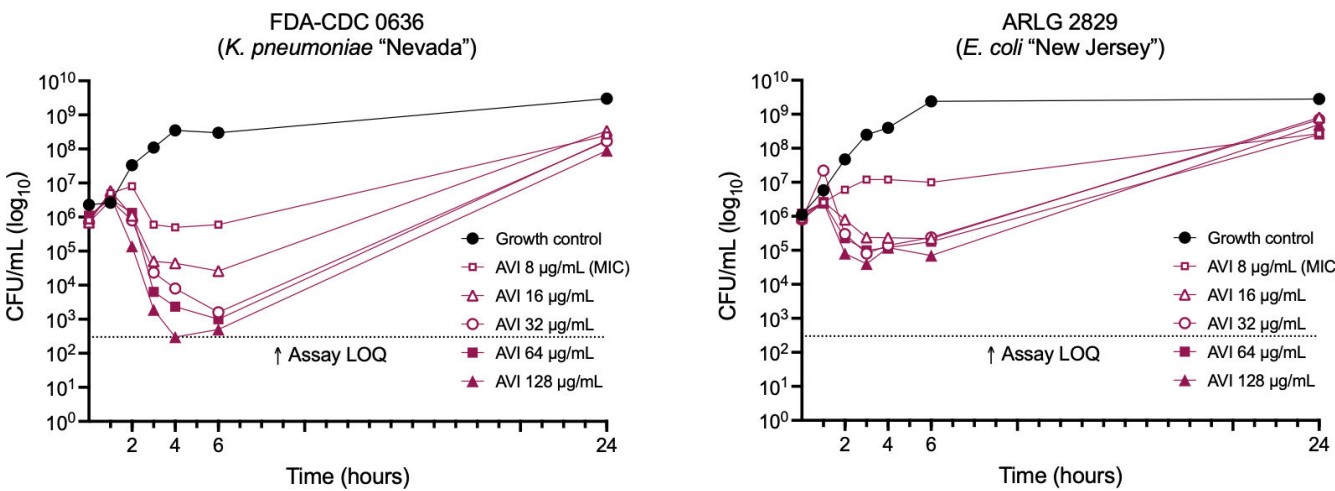

**FIG 2** Time-kill studies of *K. pneumoniae* FDA-CDC 0636 and *E. coli* ARLG 2829 grown with avibactam. LOQ, limit of quantitation.

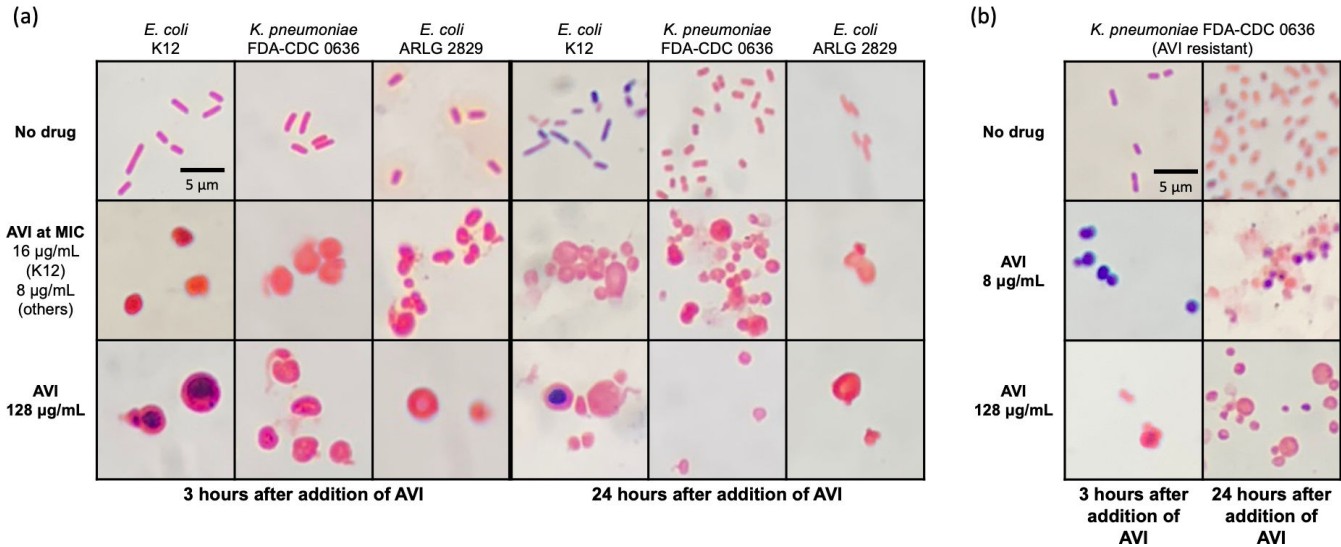

**FIG 3** Gram stain images of avibactam-susceptible cells (a) and avibactam-resistant cells (b) grown with different concentrations of avibactam, 1,000×.

antibiotic which, like AVI, exclusively targets PBP2 (23, 27), increased 16× for BIDMC 22 and >64× for BIDMC 49A, while the MIC of zidebactam, a DBO with potent PBP2-inhibiting activity (28), increased by >1,024× for both strains (Table 2). In two other cases (amoxicillin for BIDMC 22 and cefepime for BIDMC 49A), MICs increased 4×; both of these drugs exert partial PBP2 binding (29, 30). Post-AVI MICs in all others changed by no more than a single twofold doubling dilution, which is within the expected range of variability for MIC assays. The MICs of ceftazidime and ceftazidime-avibactam also remained unchanged when tested in the three AVI-resistant derivatives of *E. coli* K12 described below.

## Avibactam resistance emerges readily during drug exposure and persists in the absence of selective pressure

To determine whether regrowth of bacteria in time-kill experiments resulted from the development of heritable AVI resistance, cells were recovered after 24 hours of growth in media containing 128 µg/mL AVI, and sequential AVI MIC testing was performed on isolates subjected to daily serial subcultures on antibiotic-free media over the course of 15 days. The MICs of the three different strains on which this procedure was performed (*E. coli* K12, *K. pneumoniae* FDA-CDC 0636, and *E. coli* ARLG 2829) remained >256 µg/mL over the course of the experiment (32×–64– starting MIC).

A diffusion-based tolerance test (31, 32) was performed to determine whether tolerance to AVI was also contributing to regrowth (33). A lawn of *E. coli* K12 was incubated overnight in the presence of an AVI-impregnated disk, which was then removed and replaced with a glucose-containing disk for nutrient repletion, to allow for the regrowth of colonies from any tolerant cells that had survived in the ~20 mm zone of clearance. After a subsequent night of incubation, no colonies had appeared in the zone of clearance, indicating an absence of AVI tolerance and confirming that regrowth in time-kill studies represented true heritable resistance.

## Rapid emergence of avibactam resistance is the result of a high resistance mutation frequency, even in stringent response-deficient strains

Because resistance to PBP2-targeting drugs is thought to involve activation of the stringent response (19), time-kill studies were also performed using derivatives of *E. coli* K12 with inactivating mutations in the stringent response pathway (34) as well as the SOS response pathway (35) (Table S2). Strains were grown for 48 hours with AVI at 128 µg/mL (8×–16× MIC). At 24 hours, the ΔspoT strain had regrown to a similar density

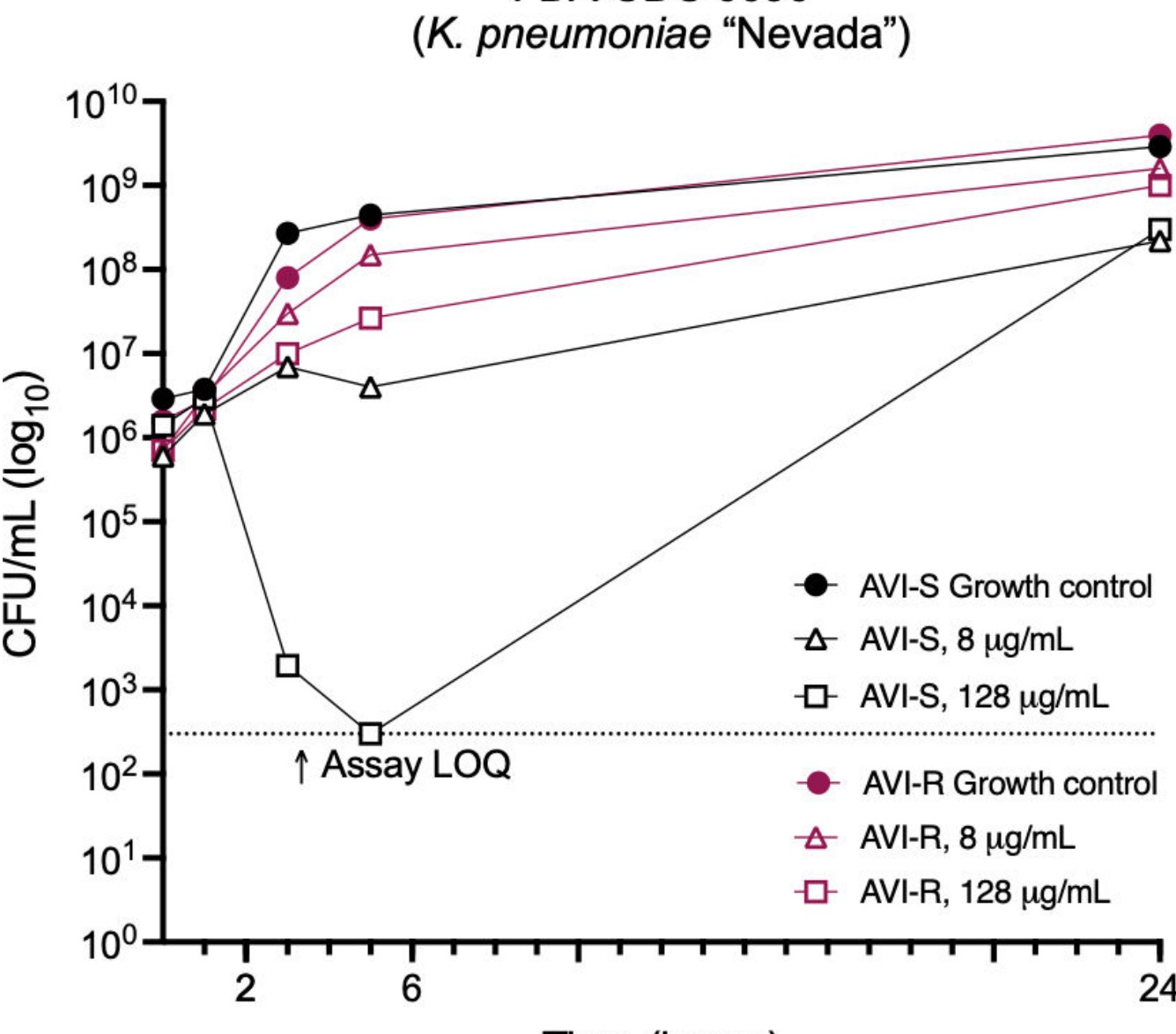

**FIG 4** Time-kill studies of *K. pneumoniae* FDA-CDC 0636 parent and avibactam-resistant derivative strains grown with avibactam. LOQ, limit of quantitation.

as the K12 parent strain, while the ΔrecA and ΔrelA strains had cell densities that were lower by 0.8 and 1.5 $\log_{10}$ CFU/mL, respectively, although both had still regrown above the starting inoculum. The most notable effect on regrowth at 24 hours was seen in the strain lacking both *relA* and *spoT*, which had 3.4 $\log_{10}$ CFU/mL fewer cells than the parent strain and 1.8 $\log_{10}$ CFU/mL fewer than its own starting inoculum. By 48 hours, all treated strains had a similar cell density (Fig. 5).

The mutation frequency for *E. coli* K12 was $8.5 \times 10^{-6}$ and $6.6 \times 10^{-6}$ at 4× and 8× MIC, respectively (Fig. 6). Mutation frequency rates for the other strains tested were similar. Interestingly, there was no statistically significant decrease in mutation frequency between K12 and strains with deletions in key stringent response genes ($P > 0.5$ by unpaired two-tailed *t*-test); indeed, the only significant differences in mutation frequency between K12 and other strains were an increase in mutation frequency at 4× MIC in a Δ*spoT* strain ($P = 0.048$) and a Δ*recA* strain ($P = 0.043$).

**TABLE 2**  Effect of AVI resistance on MICs of other drugs[a,b]

| Drug | PBP target(s) | *K. pneumoniae* BIDMC 22 | | | *E. coli* BIDMC 49A | | |
|------|---------------|-------------|--------------|-------------|-------------|--------------|-------------|
| | | Initial MIC | Post-AVI MIC | MIC change | Initial MIC | Post-AVI MIC | MIC change |
| Avibactam | 2 | 16 | 128 | **8×** | 8 | 128 | **16×** |
| Mecillinam | 2 | 0.25 | >16 | **>64×** | 0.063 | 1 | **16×** |
| Zidebactam | 2 | 0.125 | >128 | **>1,024×** | 0.125 | >128 | **>1,024×** |
| Meropenem | 2 > 4 > 3 > 1 | 0.063 | 0.063 | None | 0.016 | 0.031 | 2× |
| Amoxicillin | 4 > 2 > 3 | 8 | 32 | **4×** | 4 | 2 | −2× |
| Cefepime | 3 > 2 > 1 > 4 | 0.063 | 0.063 | None | 0.031 | 0.125 | **4×** |
| Ceftazidime | 3 > 1 | 0.25 | 0.25 | None | 0.25 | 0.25 | None |
| Aztreonam | 3 | 0.063 | 0.063 | None | 0.063 | 0.125 | 2× |

[a]PBP, penicillin-binding protein. MICs are expressed in microgram per milliliter.
[b]Bold text indicates MIC increases of >2 doubling dilutions.

## A diverse set of mutations and methylation changes underlie avibactam resistance

Illumina sequencing was performed on 2–3 AVI-R mutants each of *E. coli* K12, a K12 Δ*spoT* derivative, a K12 Δ*spoT*/Δ*relA* derivative, and NEB 5-alpha in order (i) to determine whether AVI resistance-related mutations were similar to those seen in bacteria resistant to other PBP2-targeting drugs like mecillinam and nacubactam (16, 19) and (ii) to determine whether absence of stringent and SOS response genes would result in a different mutational pattern. Sequencing reads were aligned to the *E. coli* K12 reference genome, and variants were called using Pilon (36) (Table 3). One of the K12 mutants had a 1,336 bp insertion sequence (IS2) 145 bp upstream of threonine-tRNA ligase (*thrS*), likely in the promoter region of the gene; mutations in *thrS* have previously been reported in mecillinam-resistance *E. coli* strains (16). Both K12 mutants had point mutations causing amino acid changes in *cyaA* (adenylate cyclase), another gene that has previously been implicated in mecillinam resistance (37). None of the mutations seen in the two Δ*spoT* mutants have been previously described, but both strains had the same amino acid change (A63D) in *fabR*, which encodes a transcriptional regulator that represses unsaturated fatty acid synthesis (38). In 5 of the 10 strains (one K12 mutant, both Δ*spoT* mutants, and two of three Δ*spoT*/Δ*relA* mutants), there was a large intergenic insertion between an IS5 family transposase and *oppA*, which encodes an oligopeptide ABC transporter periplasmic binding protein. OppA is the periplasmic component of an oligopeptide transport system and has also been implicated as a cause of aminoglycoside resistance, possibly because the protein plays a role in aminoglycoside uptake by the cell (39). One of the Δ*spoT*/Δ*relA* mutants had a 7 bp deletion resulting in a premature stop codon in *tolB*, the gene encoding a periplasmic protein in the Tol-Pal system, which is involved in bacterial cell division (40). A premature stop codon in *tolB* has previously been described in a nacubactam-resistant isolate (19). Interestingly, each of the three mutants of NEB 5-alpha had a 1,338 bp insertion within *cysB*, the gene encoding the transcriptional regulator CysB, which controls cysteine biosynthesis. Inactivating mutations in *cysB* have been found in the majority of clinical mecillinam-resistant isolates, potentially because they cause a lower fitness cost than other mutations conferring resistance to PBP2 inhibition, yet have rarely been reported in laboratory-selected strains (16).

The complexity and variety of AVI resistance-conferring genomic mutations prompted consideration of whether epigenetic changes, in the form of differential methylation, could also be playing a role in resistance. We thus used long-read sequencing technology to quantify methylation of sites throughout the genomes of the strains that had undergone whole-genome sequencing. 5-Methylcytosine (5mc) and N6-deoxyadenosine (6ma) methylation were predicted from Oxford Nanopore sequencing data. In total, sites of differential methylation (see "Materials and Methods" for criteria used to identify these sites) occurred in 448 different genes and 39 intergenic regions in the case of 5mc methylation, but only 11 genes and 9 intergenic regions for 6ma methylation

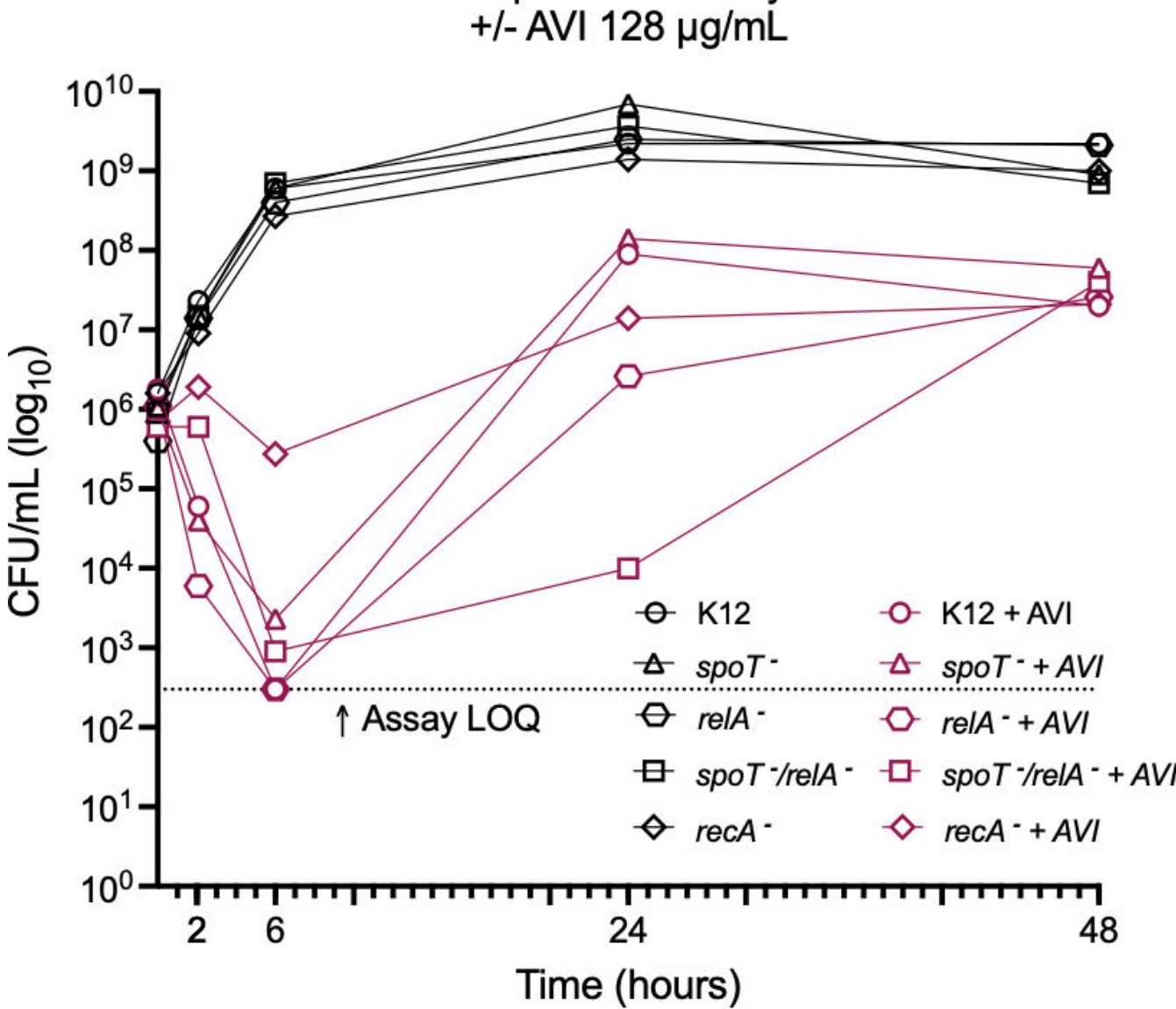

**FIG 5** Time-kill studies of *E. coli* K12 and strains with mutations in stringent response (*spoT* and *relA*) and SOS (*recA*) genes. LOQ, limit of quantitation.

(Tables S3 and S4). Sites of differential 6ma methylation were unevenly distributed, with 10 of 35 sites of differential methylation occurring in the putative outer membrane porin gene *nmpC* or the adjacent intergenic region; in all cases, these represented a decrease in methylation in AVI-resistant mutants derived from NEB 5-alpha. Interestingly, this gene is located near *rusA*, which was the most represented gene in 5mc methylation differences (see below). Overall, NEB 5-alpha was greatly overrepresented in 6ma methylation, with 75 sites of differential methylation occurring across the three NEB 5-alpha mutants and only 16 in all other mutants combined.

Sites of 5mc differential methylation were spread more evenly across strain backgrounds, with many sites occurring in multiple strain backgrounds (Table S4). The gene with the greatest number of 5mc differences was *rusA*, which encodes an endonuclease that resolves Holliday Junction intermediates created during DNA repair by homologous recombination (41). In two of the NEB 5-alpha mutants, *rusA* 5mc methylation was decreased at five different sites, while in all three ΔspoT/ΔrelA mutants, methylation in this gene was increased at a separate site. *RuvC*, another such "resolvase" (42), was also

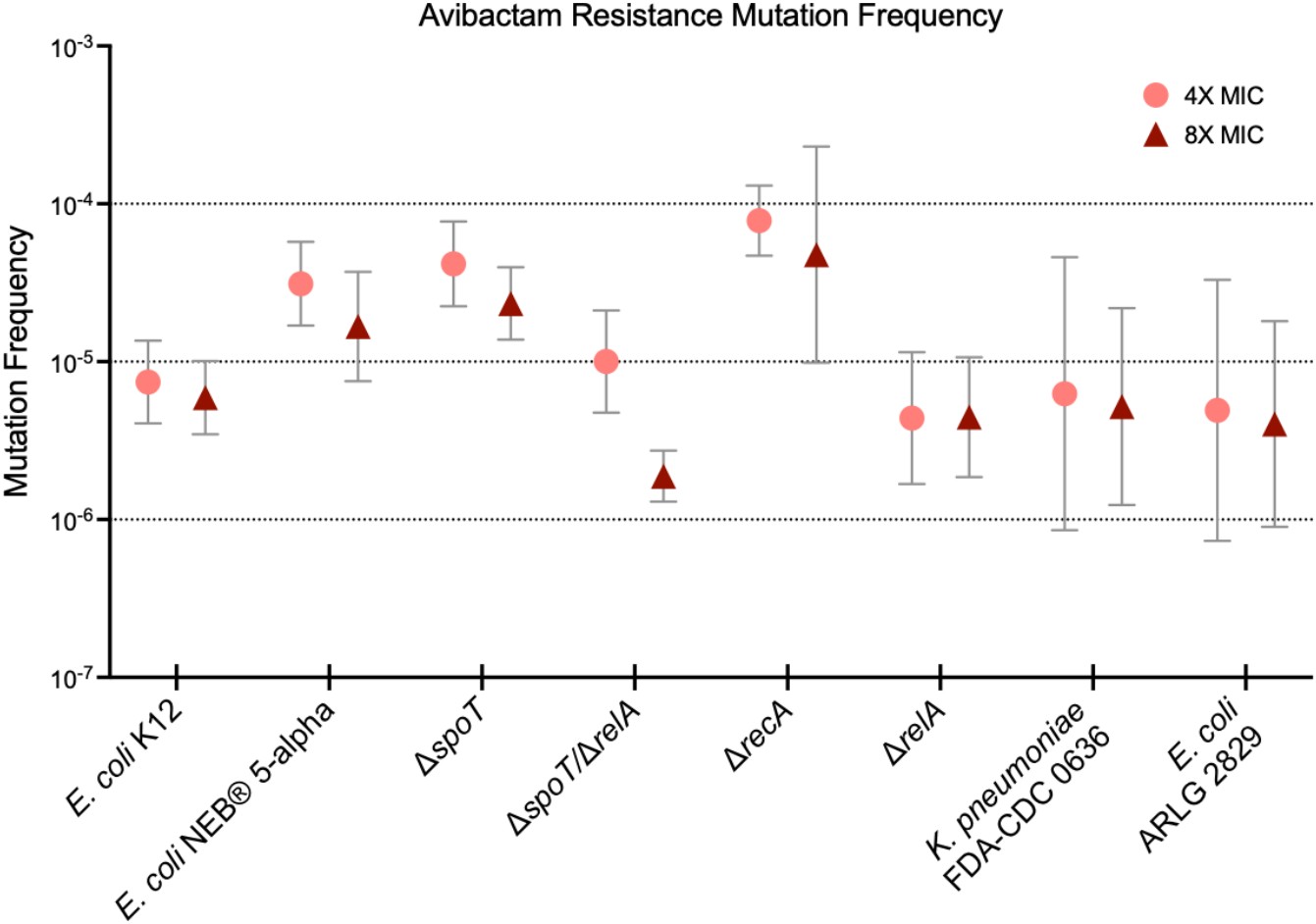

**FIG 6** Avibactam resistance mutation frequency at multiples of the AVI MIC. Symbols and bars indicate the geometric mean and SD of three replicates.

differentially methylated in several samples, with decreased methylation in two NEB 5-alpha mutants and one K12 mutant and increased methylation in two Δ*spoT*/Δ*relA* mutants. While these data do not provide information on whether methylation at these sites resulted in increased or decreased gene expression, the predominance of genes involved in DNA repair is notable. Many of the genes that have previously been implicated in resistance to PBP2-targeting drugs were also differentially methylated, including several tRNA ligases (*alaS*, *asnS*, *proS*, *serS*, and *thrS*), as well as *cyaA* (adenylate cyclase), *cysE*, *ubiX* (16), *arcA*, *cydA*, and *tolB* (19).

### Avibactam resistance confers a modest *in vitro* fitness cost

In a growth rate assay performed to assess for potential fitness costs of AVI resistance, two AVI-resistant mutant derivatives of *E. coli* K12 showed increased lag time relative to the parent strain but did not have a statistically significant decrease in growth rate (Fig. 7). Resistance to PBP2-targeting drugs involves compensatory mechanisms that allow survival and replication in the presence of PBP2 inhibition, but the abnormal morphology of cells grown in the presence of these drugs (Fig. 3b) suggests the possibility of a further fitness cost during drug exposure. To evaluate this possibility, we also tested the growth fitness of the two AVI-resistant isolates in the presence of a sub-MIC AVI concentration (128 µg/mL). Both strains showed an increase in lag time when grown with AVI, but only one of the strains (mutant #2) demonstrated a significantly decreased growth rate. Interestingly, both of these strains have mutations in the gene encoding adenylate cyclase, but mutant #1 also has an insertion sequence 145 bp upstream of

**TABLE 3** Genetic changes identified in avibactam-resistant strains[a,b]

| Position in K12 | Ref base | Variant | K12-1 | K12-2 | ΔspoT-1 | ΔspoT-2 | ΔspoT/ΔrelA-1 | ΔspoT/ΔrelA-2 | ΔspoT/ΔrelA-3 | NEB5a-1 | NEB5a-2 | NEB5a-3 | Location | Notes | Annotation (gene products in bold have previously been reported to confer resistance to PBP2-targeting drugs) |
|---|---|---|---|---|---|---|---|---|---|---|---|---|---|---|---|
| 125,154 | G | A | | | | | | ▪ | | | | | Coding | G713D | Pyruvate dehydrogenase E1 component (AceE) |
| 144,672 | C | A | ▪ | | | | | | | | | | Coding | Synonymous | Putative phosphotransferase enzyme IIA component (YadI) |
| 600,627 | A | T | | | | | | | | ▪ | | | Coding | Synonymous | Cation efflux system protein (CusA) |
| 778,320 | T | 7 bp del. | | | | | ▪ | | | | ▪ | | Coding | | **Tol-Pal system periplasmic protein (TolB)** (18) |
| 891,652 | C | 1,338 bp ins. | | | | ▪ | | | | | ▪ | | Coding | | NADPH-dependent nitro/quinone reductase (NfsA) |
| 1,207,805 | G | 3,527 bp ins. | | | | | ▪ | | | | | | Coding | | Hypothetical protein (StfP) |
| 1,209,618 | A | 3,455 bp ins. | | | | | | | | | ▪ | | Coding | | Hypothetical protein (StfE) |
| 1,209,618 | A | 3,637 bp ins. | | | | | ▪ | | | | ▪ | | Coding | | Hypothetical protein (StfE) |
| 1,216,333 | T | G | | | | | | | | | ▪ | | Coding | I89R | Putative two-component system connector protein (YmgA) |
| 1,300,695 | T | 1,629 bp ins. | ▪ | | ▪ | ▪ | | | | | | | Intergenic | 148 bp downstream of insH21; 487 bp upstream of oppA | IS5 family transposase and trans-activator (InsH21); oligopeptide ABC transporter periplasmic binding protein (OppA) |
| 1,300,697 | A | 1,142–2,590 bp ins. | | | | ▪ | | | | | | | Intergenic | 150 bp downstream of insH21; 485 bp upstream of oppA | IS5 family transposase and trans-activator (InsH21); oligopeptide ABC transporter periplasmic binding protein (OppA) |
| 1,333,990 | A | 1,338 bp ins. | | | | | | | | ▪ | | | Coding | | **DNA-binding transcriptional dual regulator (CysB)** (16, 18) |
| 1,334,312 | A | 1,338 bp ins. | | | | | | | | | | ▪ | Coding | | **DNA-binding transcriptional dual regulator (CysB)** |
| 1,334,512 | A | 1,338 bp ins. | | | | | | | | | ▪ | ▪ | Coding | | **DNA-binding transcriptional dual regulator (CysB)** |
| 1,563,289 | T | C | | | ▪ | | | | | | | | Intergenic | 213 bp upstream of ddpX; 45 bp downstream of dosP | D-alanyl-D-alanine dipeptidase (DpX); oxygen-sensing c-di-GMP phosphodiesterase (DosP) |
| 1,749,599 | G | 19,285 bp del. | | | | | | | | | | ▪ | | Deletion of large segment containing multiple genes | Putative cytochrome (YdhU; partial deletion), putative 4Fe-4S ferredoxin-like protein (YdhX), uncharacterized protein (YdhW), putative oxidoreductase (YdhV), putative 4Fe-4S ferredoxin-like protein (YdhY), fumarase D (FumD), protein (YnhH), pyruvate kinase 1 (PykF), murein lipoprotein (lpp), L,D-transpeptidase (LdtE), sulfur carrier protein (SufE), |

*(Continued on next page)*

**TABLE 3** Genetic changes identified in avibactam-resistant strains[a,b] (*Continued*)

| Position in K12 | Ref base | Variant | K12-1 | K12-2 | ΔspoT-1 | ΔspoT-2 | ΔspoT/ΔrelA-1 | ΔspoT/ΔrelA-2 | ΔspoT/ΔrelA-3 | NEB5a-1 | NEB5a-2 | NEB5a-3 | Location | Notes | Annotation (gene products in bold have previously been reported to confer resistance to PBP2-targeting drugs) |
|---|---|---|---|---|---|---|---|---|---|---|---|---|---|---|---|
| | | | | | | | | | | | | ■ | | | L-cysteine desulfurase (SufS), Fe-S cluster scaffold complex subunits (SufD, SufC, and SufB), iron-sulfur cluster insertion protein (SufA), small regulatory RNA (RydB), uncharacterized protein (YdiH), 1,4-dihydroxy-2-naphthoyl-CoA hydrolase (MenI), and putative FAD-linked oxidoreductase (YdiJ) |
| 1,802,720 | C | 1,336 bp ins. | ■ | | | | | | | | | | Intergenic | 150 bp upstream of *thrS*; 374 bp upstream of *arpB* | **Threonine tRNA ligase (ThrS)** (16); pseudo gene (ArpB) |
| 1,961,820 | T | 14 bp del. | | | | | | | | | ■ | | Intergenic | 25 bp downstream of *argS*; 17 bp downstream of *yecV* | **Arginine-tRNA ligase (ArgS)** (16); protein YecV |
| 1,978,494 | A | 632 bp ins. | ■ | | | | | | | | | | Intergenic | 297 bp upstream of *flhD*; 24 bp downstream of *insB5* | DNA-binding transcriptional dual regulator (FlhD); IS1 family protein (InsB) |
| 2,535,136 | G | 51 bp del. | | | | | ■ | | | | | | Coding | | PTS enzyme I |
| 2,868,929 | C | 1,112 bp ins. | | | | | | | | ■ | | | Coding | | Protein-L-isoaspartate O-methyltransferase (pcm) |
| 2,999,673 | A | 1 bp del. | | | | | | ■ | | | | | Coding | | Amidase activator (ActS) |
| 3,277,257 | C | 7 bp ins. | | | | | | | | ■ | | | Coding | | Antitoxin (PrlF) |
| 3,423,530 | C | 245 bp del. | ■ | | | | | | | | | | RNA | | Deletion of 5S ribosomal RNA (*rrfF*; partial deletion), tRNA-Thr(GGU) (*thrV*), 5S ribosomal RNA (*rrfD*; partial deletion) |
| 3,969,169 | T | 1 bp del. | | | | | | | ■ | | | | Coding | | Enterobacterial common antigen polysaccharide co-polymerase (wzzE) |
| 3,992,052 | T | A | | | | | | | | | | | Coding | D300E | **Adenylate cyclase (CyaA)** (36) |
| 3,993,380 | A | C | ■ | | | | | | | | | | Coding | Q743P | **Adenylate cyclase (CyaA)** |
| 4,104,816 | C | T | | | | ■ | | | | | | | Coding | E54K | Sensor histidine kinase (CpxA) |
| 4,161,254 | C | A | | | | | ■ | | | | | | Coding | A63D | DNA-binding transcriptional repressor (FabR) |
| 4,475,509 | G | A | | | | | | | | ■ | | | Coding | V25I | Uncharacterized protein (YjgL) |
| 4,612,472 | A | C | | | | | | | ■ | | | | Coding | Q21P | Putative patatin-like phospholipase (YjjU) |

[a]Boxes indicate variants within 2 kb of each other.

[b]Gray shading indicates the presence of the mutation in a given strain.

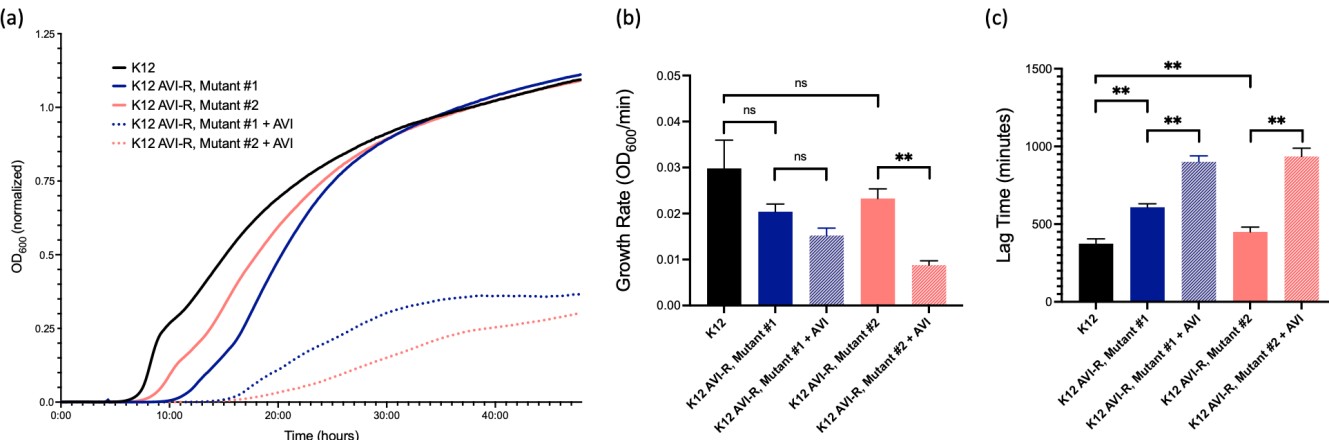

**FIG 7** Growth fitness assay demonstrating growth over time of *E. coli* K12 and AVI-R mutant derivative strains grown with and without AVI 128 µg/mL. (a) Growth curves. Data represent the mean of three biological replicates. Readings are normalized to media-only wells. (b and c) Growth rates and lag time. Measurements in Fig. 7b and c were calculated using GrowthRates 6.2.1 (Bellingham Research Institute). Mean values with SD are shown. **, $P < 0.01$; ns, nonsignificant via paired two-tailed *t*-test.

*thrS* (threonine-tRNA ligase), potentially in the promoter region (Table 3). These results suggest that different collections of AVI-resistance mutations may confer differential fitness costs.

## Avibactam appears to have greater *in vivo* efficacy in an immunocompetent mouse model compared to a neutropenic model

In the neutropenic thigh infection model, mice infected with *K. pneumoniae* FDA-CDC 0636 and treated with 250 mg/kg AVI every 8 hours for three doses had a bacterial burden at 24 hours that was 0.5 $\log_{10}$ CFU/thigh lower than in mice treated with saline (Mann-Whitney $U = 0$; $P = 0.029$; Fig. 8). To evaluate the possible effect of an intact innate immune response on AVI activity, AVI was also evaluated in a non-neutropenic thigh infection model. In immunocompetent mice, AVI had greater activity, with treated mice showing a bacterial burden 1.7 $\log_{10}$ CFU/thigh lower than saline-treated mice (Mann-Whitney $U = 0.5$; $P = 0.016$; Fig. 8). The AVI MIC of bacteria recovered from

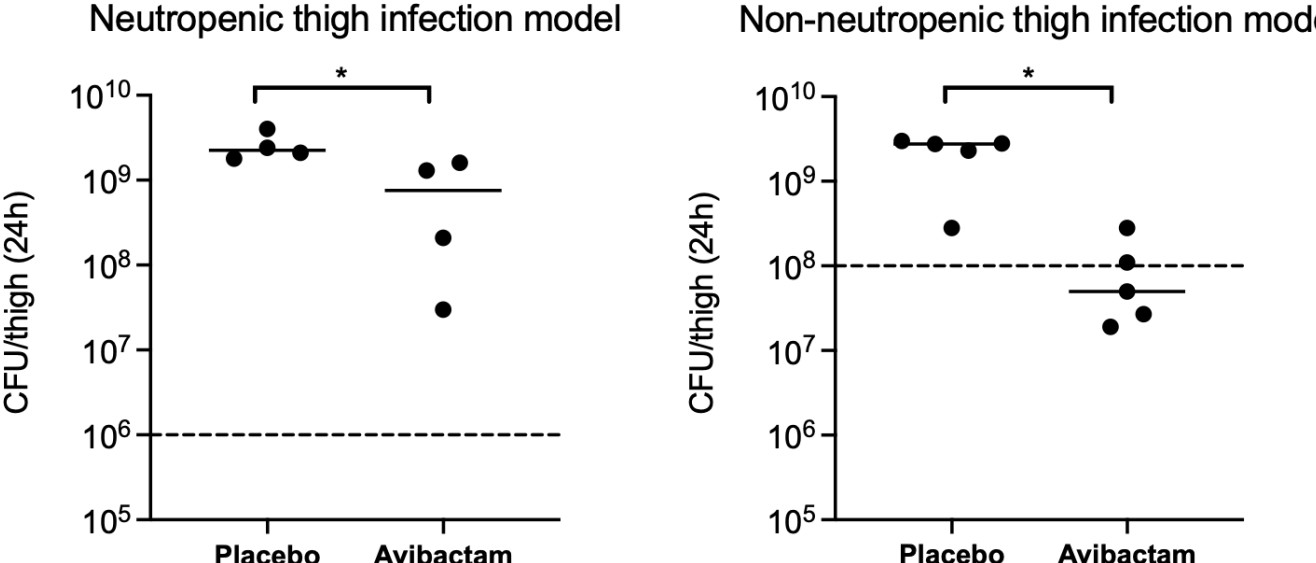

**FIG 8** Colony counts of *K. pneumoniae* FDA-CDC 0636 ("Nevada" strain) at 24 hours in mice treated with placebo or avibactam. *, $P < 0.05$ via Mann-Whitney *U* test. Dashed lines indicate starting inoculum.

the thighs of the immunocompetent mice treated with AVI was unchanged from the baseline MIC of the strain (8 µg/mL), suggesting that resistance had not emerged during treatment.

## DISCUSSION

Bacterial resistance to drugs that inhibit PBP2 occurs through mutations in a remarkably large and heterogeneous collection of genes, whose commonality appears to be involved in the stringent response pathway. Upregulation of this bacterial stress response results in the ability of bacteria to survive and replicate in spite of PBP inhibition. This unusual approach to resistance has been studied to date primarily in the context of mecillinam, the sole β-lactam antibiotic that targets only PBP2 (43). However, DBO β-lactamase inhibitors also exert their direct antimicrobial activity through inhibition of PBP2, and mutations similar to those identified in mecillinam-resistant strains have been identified in bacteria resistant to the DBO nacubactam (19). In this work, we have contributed to the understanding of resistance to PBP2-inhibiting drugs by thoroughly characterizing the development of resistance to the direct activity of the DBO avibactam.

We found that resistance to AVI emerged readily upon exposure to the drug at concentrations as much as 16 times the MIC. This phenomenon first became apparent in time-kill studies, where bacterial regrowth occurred reliably by 24 hours (Fig. 2 and 6). The explanation for the discrepancy between inhibition of growth in the MIC assay and failure of inhibition in the time-kill assay can be understood by considering the baseline AVI resistance frequency of $1.97 \times 10^{-6}$ to $8.15 \times 10^{-5}$ among the strains we tested. The starting bacterial inoculum in time-kill studies is ~$10^8$ cells ($10^6$ CFU/mL × 10 mL volume), whereas, in the 384-well plate MIC method, it is $2.5 \times 10^4$ ($5 \times 10^5$ CFU/mL × 0.05 mL volume), thus at least one AVI-resistant cell is almost certain to occur in a time-kill study, but not in an MIC assay. Mecillinam is also characterized by a high mutation frequency, such that it is only used to treat bladder infections, where its concentration at extremely high levels in urine limits the emergence of resistance (16, 44). AVI resistance persisted over more than 2 weeks of serial subcultures in the absence of antibiotic pressure, demonstrating that cells had developed true heritable resistance, and there was no evidence of tolerance when this was tested directly.

As expected, resistance to AVI conferred resistance to other exclusively PBP2-targeting drugs but not to β-lactam antibiotics with different or additional PBP targets (Table 2). The morphological appearance of AVI-resistant cells grown in the presence of AVI was consistent with the enlarged, round forms seen in cells resistant to other PBP2-targeting drugs, including mecillinam (24, 45) and nacubactam (19) (Fig. 3b). The fact that AVI-resistant and AVI-susceptible cells take on the same distorted appearance in the presence of AVI is reflective of the remarkable approach taken by bacteria in developing resistance to PBP2-inhibiting drugs. In the great majority of cases, resistance to PBP2-targeting drugs in *Enterobacterales* occurs via the emergence of one or more of a large number of apparently compensatory mutations, which allow bacteria to survive and replicate in the presence of PBP2 inhibition, rather than preventing PBP2 inhibition outright (16). This large mutational target is believed to confer resistance to PBP2 inhibitors by upregulation of the stringent response (46). Activation of the *ftsAQZ* operon and the resultant ability of bacterial cells to survive and replicate as the enlarged, rounded forms generated by PBP2 inhibition has been proposed as the final step leading from stringent response activation to resistance to PBP2 inhibition (19). However, the process—and the role of the many different mutations identified in isolates resistant to PBP2 inhibition—remains incompletely understood. In two AVI-resistant mutants derived from *E. coli* K12, we found non-synonymous point mutations in *cyaA*, the gene encoding adenylate cyclase, which has previously been described as a cause of mecillinam resistance (37, 47), potentially mediated by effects on lipopolysaccharide synthesis (47). One of the strains also had an insertion sequence in the *thrS* gene encoding threonine-tRNA ligase. Mutations in tRNA synthetases are among the most common

genetic changes identified in bacteria resistant to PBP2-inhibiting drugs (16, 43) and are believed to simulate amino acid starvation, resulting in a stimulus for upregulation of the stringent response (19). To evaluate the emergence of AVI resistance in a different *E. coli* K12 strain background, we sequenced three AVI-resistant mutants of NEB 5-alpha, a derivative of DH5α (48). Interestingly, all three of these isolates had an insertion sequence within *cysB*. Inactivation of CysB, a positive regulator of cysteine biosynthesis, is identified frequently in clinical mecillinam-resistant *E. coli* isolates but is rarely observed in standard laboratory-selected mutants, potentially due to increased growth fitness in urine (16). CysB inactivation is believed to confer mecillinam resistance through a pathway that is independent of the stringent response but has similar downstream effects, ultimately rendering PBP2 apparently inessential (49). Although it is not clear why this mutation would be preferentially selected in the NEB 5-alpha strain background, this finding suggests that differences in strain characteristics may have important impacts on the type of mutations that emerge to PBP2-targeting drugs and, in turn, on the fitness and potential clinical importance of these isolates. Because sequencing for each parent strain was performed on isolated colonies generated from a single starting culture, the detection of multiple strains with the same mutation could reflect the emergence of resistant mutants at one or more potential time points: (i) pre-existing mutations present at a low level within the parent strain population and selected under AVI pressure, (ii) early emergence of a mutant clone giving rise to multiple colonies, or (iii) convergent evolution of the same mutation in multiple different cells. Future experiments, potentially using methods in development such as single-cell whole-genome sequencing, may be able to clarify the time course of the emergence of resistance.

The fact that mutations causing upregulation of the stringent response predominate in laboratory-selected strains resistant to PBP2 inhibition raised the question of whether, and how, strains deficient in key genes in the stringent response pathway might develop AVI resistance. Surprisingly, we saw no significant decrease in mutation frequency in strains lacking *spoT*, *relA*, or both (Fig. 5). However, the genes altered in AVI-R mutants of these strains were, with one exception (a seven-base pair deletion in *tolB* in one of the ΔspoT/ΔrelA double mutants [19]), genes that have not previously been described in bacteria resistant to PBP2-targeting drugs and, as expected, are not part of the stringent response pathway. TolB is a component of an alternate path that may result in activation of the *ftsQAZ* operon (19), but to our knowledge, the other genes in which we identified mutations are not known to participate in this process. Our analysis of differences in methylation between AVI-resistant mutants and their parent strains highlighted several of the same genes in which mutations are frequently found in PBP2 inhibitor resistance, as well as several genes not previously implicated in resistance to these drugs. Our cumulative data underscore the strikingly complex and multifarious mechanisms by which bacteria develop resistance to PBP2-targeting drugs, although the reason for this unusual approach to resistance remains opaque.

Our study has certain limitations. We have not yet induced mutations in the novel genes we identified to confirm that their inactivation confers AVI resistance; this will be an important step in future work. In addition, our analysis of methylation data does not provide information on whether genes with increases in methylation are silenced, and a more detailed investigation of methylation patterns and gene expression levels in the future will help to elucidate the role of methylation in resistance to PBP2-targeting drugs.

The study of resistance to PBP2-targeting drugs has long been of interest in the context of bacterial physiology and cell wall morphology. However, with the development of β-lactam/β-lactamase inhibitor combinations in which the β-lactamase inhibitor is a DBO compound with potent direct antimicrobial activity (e.g., zidebactam and nacubactam), an understanding of resistance to PBP2 inhibition will be increasingly clinically relevant. The study of resistance to these combinations will have to take into account the roles of the β-lactam drug and the β-lactamase inhibitor function of the DBO, in addition to direct DBO activity. Of note, however, the activity of these combinations against MBL-producing strains relies primarily on the inhibition of PBP2 by the DBO,

as DBOs do not inhibit MBLs and therefore cannot protect their partner β-lactam drug against MBL-mediated hydrolysis (50).

A key question about the activity of combinations of β-lactams with high-potency DBOs is how resistance will emerge *in vivo*. In the present work, we did observe a fitness cost, in terms of lag time but not growth rate, in AVI-resistant isolates (Fig. 7), which might suggest that AVI-resistant isolates would be less likely to survive in the more exacting host environment. The fact that AVI appeared more active in an immunocompetent mouse model compared to an otherwise identical neutropenic model (Fig. 8) and that bacteria recovered from immunocompetent mice after treatment with AVI did not show any increase in AVI MIC provides support for the idea that host responses may reduce the likelihood of emergence of resistance. Indeed, Ulloa et al. have noted that components of the innate immune system exert synergistic activity with AVI against MBL-producing *K. pneumoniae* (51). However, the large mutational target leading to AVI resistance suggests that bacteria exposed to DBOs *in vivo* may preferentially select options from the "menu" of mutations that allow for improved survival in the presence of AVI (Fig. 7) and under various selective pressures exerted by the host.

How the presence of a paired β-lactam drug and the higher potency of new DBOs will affect the ability of bacteria to develop resistance *in vivo* remains to be learned. Our future work will involve the study of resistance to combinations of new, highly potent DBOs with β-lactams using both *in vitro* and preclinical models. These new agents offer an enticingly broad spectrum of activity against MDR gram-negative bacteria, but the reliance of this activity on direct inhibition of PBP2 by DBOs will require ongoing studies of mechanisms of resistance in order to preserve their activity in the ongoing fight against MDR gram-negative pathogens.

## MATERIALS AND METHODS

### Bacterial strains

Bacteria were obtained from the following sources (Table S1 and S2): the U.S. FDA-CDC (Centers for Disease Control and Prevention) Antimicrobial Resistance Isolate Bank (48 isolates), the Antibiotic Resistance Leadership Group Laboratory Center Virtual Repository (one isolate), the carbapenem-resistant *Enterobacteriaceae* genome initiative at the Broad Institute in Cambridge, MA (21 isolates) (52, 53), New England BioLabs (one isolate), and the Coli Genetic Stock Center (four isolates). *E. coli* ATCC 25922, *Staphylococcus aureus* ATCC 29213, *K. pneumoniae* ATCC 700603, *K. pneumoniae* ATCC 13883, *K. pneumoniae* ATCC BAA-1705, and *Pseudomonas aeruginosa* ATCC 27853 were obtained from the American Type Culture Collection (Manassas, VA). All strains were colony purified, minimally passaged, and stored at −80°C in tryptic soy broth (TSA; BD Diagnostics, Franklin Lakes, NJ) with 50% glycerol (Sigma-Aldrich, St. Louis, MO) prior to use in this study.

### Antimicrobial agents

Ceftazidime and cefepime were obtained from Chem Impex International (Wood Dale, IL). Avibactam was obtained from MedChemExpress (Monmouth Junction, NJ). Clavulanate was obtained from Sigma-Aldrich (St. Louis, MO). Amoxicillin was obtained from Alfa Aesar (Tewksbury, MA). Mecillinam (amdinocillin) was obtained from Research Products International (Mt. Prospect, IL). Meropenem was obtained from Ark Pharm, (Libertyville, IL). Aztreonam was obtained from MP Biomedicals (Solon, OH). Antimicrobial stock solutions used for the DDM were dissolved in water or in dimethyl sulfoxide according to Clinical and Laboratory Standards Institute (CLSI) recommendations (54); 0.3% polysorbate 20 (P-20; Sigma-Aldrich, St. Louis, MO) was added to the water used for these solutions as required by the HP D300 digital dispenser instrument (HP, Inc., Palo Alto, CA) for proper fluid handling. As recommended by CLSI, anhydrous sodium carbonate at 10% ceftazidime weight was added to the ceftazidime stock solution. The antibiotic stock solutions used for time-kill studies and agar dilution plates were

dissolved in water, with the addition of anhydrous sodium carbonate at 10% weight/ weight for ceftazidime. All antibiotic stock solutions were quality control (QC) tested with *E. coli* ATCC 25922, *S. aureus* ATCC 29213, *K. pneumoniae* ATCC 700603, and/or *K. pneumoniae* ATCC BAA-1705 using the D300 dispensing method described below (for stocks to be used for checkerboard arrays) or standard broth microdilution using the direct colony suspension method (55) (for stocks to be used for time-kill experiments) prior to use in synergy studies. Stocks were used only if they produced an MIC result within the accepted QC range according to CLSI guidelines (54). Because the MIC of avibactam for *K. pneumoniae* FDA-CDC 0636 was noted to be consistent at 8 µg/mL, this strain was used as an alternate QC organism to reduce the variables involved in QC of avibactam in combination with ceftazidime. Antimicrobials were stored as aliquots at −20°C and discarded after a single use, except for ceftazidime, which was either prepared fresh on the day of use or stored at −80°C.

## MIC and checkerboard array synergy testing

DDM MIC testing was performed using the HP D300 digital dispenser (HP, Inc., Palo Alto, CA, USA). This method, which was previously described by our laboratory, has been shown to have high precision and accuracy; the CDC employs the same method (using 96-well plates) for testing the combination of aztreonam and avibactam (56– 58). Bacterial inocula were adjusted to a McFarland reading of 0.5 and diluted 1:300 in cation-adjusted Mueller Hinton broth (CAMHB), resulting in a suspension of ~5 × $10^5$ CFU/mL, and 50 µL of this suspension was added to wells in flat-bottomed, untreated 384-well polystyrene plates (Greiner Bio-One, Monroe, NC, USA) using a multichannel pipette. Antimicrobial stock solutions were dispensed by the D300 into wells before or immediately after the addition of bacterial suspensions. Plates were incubated at 35°C– 37°C in ambient air for 16–20 hours. After incubation, bacterial growth was quantified by measurement of $OD_{600}$ using a Tecan Infinite M1000 Pro microplate reader (Tecan, Morrisville, NC, USA). An $OD_{600}$ reading of >0.08 (approximately twice typical background readings in wells containing broth alone) was considered indicative of bacterial growth; this cutoff correlated with inhibition of growth by visual assessment.

## Time-kill studies

Antibiotic stocks for time-kill studies were prepared as described above and diluted in 10 mL of CAMHB in 25 by 150 mm glass round-bottom tubes to the desired starting concentrations. The starting inoculum for time-kill studies was prepared by adding 100 µL of a 1.0 McFarland standard suspension of colonies from an overnight plate to each of the tubes. A growth control and a negative (sterility) control tube were prepared in parallel with each experiment. Cultures were incubated on a shaker in ambient air at 35°C–37°C. Aliquots from the culture were removed at serial time points, and a 10-fold dilution series was prepared in 0.9% sodium chloride. A 10 µL drop from each dilution was transferred to a Mueller-Hinton agar plate (ThermoFisher, Waltham, MA) and incubated overnight in ambient air at 35°C–37°C (59). For countable drops (drops containing 3–30 colonies), the cell density of the sample was calculated; if more than one dilution for a given sample was countable, the cell density of the two dilutions was averaged. If no drops were countable, the counts for consecutive drops above and below the countable range were averaged. The lower limit of quantitation was 300 CFU/mL.

## Persistence of avibactam resistance

Following growth in CAMHB with either no drug or 128 µg/mL AVI for 24 hours as in the time-kill method, liquid cultures were transferred to 15 mL conical centrifuge tubes and centrifuged at 5,000 g for 10 minutes at 24°C. The supernatant from each tube was poured off, and cells were resuspended in fresh CAMHB, then diluted to a 0.5 McFarland standard in 0.9% sodium chloride. AVI MIC testing was performed on these suspensions using the DDM described above; in addition, bacteria from each suspension

were isolation streaked onto a TSA/5% sheep blood agar plate (ThermoFisher). Following overnight incubation in ambient air at 35°C, isolated colonies from these plates were used to perform AVI MIC testing and for isolation streaking onto new blood agar plates. This procedure was repeated for 15 days, with MIC testing performed at seven points within this time frame. Three replicates of the entire procedure were performed with each strain.

## Mutation frequency analysis

Agar dilution plates were prepared by adding one part of a 10× antibiotic concentration to nine parts molten 1.5% Bacto agar (Becton, Dickinson and Company, Sparks, MD, USA) containing CAMHB in order to make plates containing AVI at four and eight times each strain's MIC. Bacterial strains were streaked onto TSA/5% sheep blood agar plates and incubated overnight in ambient air at 37°C. A single colony of each strain was added to 10 mL of CAMHB in a 25 by 150 mm glass round bottom tube and incubated overnight with shaking in ambient air at 37°C. A 150 µL aliquot was then removed from each culture tube and used to prepare a 1:10 dilution in 0.9% NaCl. A 10 µL spot from each dilution was placed on the agar plates with a multichannel pipette, left to dry, and incubated overnight or until visible colonies were apparent. Colonies were counted as described above, and the ratio of CFU at each antibiotic concentration to CFU in the non-antibiotic-containing plate was calculated.

## Tolerance assay

Bacterial tolerance was assessed using the TDtest described by Gefen et al. (31). AVI-containing disks were prepared by applying 10 µL of an 8 mg/mL stock solution of AVI (total quantity 80 µg; determined through initial assays to produce a zone of ~20 mm with *E. coli* K12) to a diffusion disk (BD Diagnostics, Franklin Lakes, NJ). Glucose disks were prepared by applying 5 µL of a filter-sterilized solution of 40% glucose to a disk. A 0.5 McFarland bacterial inoculum was prepared and spread onto Mueller-Hinton agar plates using Dacron swabs to create a bacterial lawn. An AVI-containing disk was placed onto the plate, which was incubated at 37°C overnight. The avibactam disk was then removed and replaced with a glucose-containing disk and again incubated overnight. Tolerance was assessed by visually inspecting the plates for bacterial growth within the zone of inhibition after the second night of incubation.

## Generation of AVI-resistant mutants for growth rate assay and sequencing

Parent strains were isolation streaked onto TSA/5% sheep blood agar plates and incubated overnight at 35°C–37°C. The following day, liquid cultures were set up from a single colony from each of the strains in 5 mL of CAMHB and incubated overnight with shaking. Cultures were then diluted to a 1.0 McFarland standard (~3 × 10⁸ CFU/mL) in saline, spread with beads on Mueller Hinton agar plates containing AVI at 128 µg/mL, and incubated overnight. The next day, three individual colonies from each plate were separately isolation streaked onto plates containing AVI at 128 µg/mL. When colonies of different morphologies (typically different sizes) appeared on the original plate, colonies of different sizes were selected to increase the likelihood of genetic diversity. Bacterial growth from these plates was then frozen in glycerol stocks at −80°C.

## Growth rate assay

*E. coli* K12 and 2 AVI-resistant mutant derivatives were streaked onto agar plates; for the AVI-resistant strains, the agar contained AVI at 128 µg/mL. Plates were incubated overnight, and bacterial inocula of approximately 1,000 CFU/mL were prepared in CAMHB from growth on the plates the following day, and 100 µL was added to wells in a 96-well plate, for a total of approximately 100 cells per well. AVI at 128 µg/mL was added to select wells using the DDM. Twelve wells were prepared for each condition (*E. coli* K12 without antibiotic and two resistant mutants with and without AVI); the remaining

wells contained CAMHB alone. The plate was incubated at 37°C for 48 hours in the Tecan Infinite M1000 Pro microplate reader, with automatic $OD_{600}$ readings obtained every 10 minutes following 10 seconds of orbital shaking. Three biological replicates of the assay were performed. The data were analyzed using GrowthRates 6.2.1 (Bellingham Research Institute) (60).

## Isolation of genomic DNA

AVI-resistant mutants were generated from *E. coli* K12, two K12-derived strains with mutations in stringent response pathway genes (Δ*spoT*, Δ*spoT*/Δ*relA*), and NEB 5-alpha, a DH5α derivative electrocompetent cloning strain in which the SOS *recA* gene is inactivated (48). To isolate genomic DNA, AVI-resistant strains were streaked from frozen stocks onto Mueller Hinton agar plates containing AVI at 128 µg/mL, and parent (AVI-susceptible) strains were streaked onto TSA/5% sheep blood agar plates and incubated overnight. Colonies from each plate were added to glass tubes containing 13 mL of CAMHB either with (AVI-R strains) or without (parent strains) 128 µg/mL AVI and incubated overnight with shaking. The following day, DNA extraction was performed using QIAGEN Genomic-tip 100/G gravity flow anion-exchange tips (QIAGEN, Germantown, MD) and QIAGEN buffers according to kit instructions. In brief, cells were pelleted by spinning culture in 15 mL conical tubes at 3,000–5,000 g for 5–10 minutes at 21°C. The supernatant was then discarded, and cells were resuspended in 3.5 mL Buffer B1 with 200 µg/mL RNase A (Monarch, New England Biolabs, Ipswich, MA) and vortexed thoroughly. Eighty microliters of 100 mg/mL lysozyme (ThermoFisher) and 100 µL of 20 mg/mL proteinase K (ThermoFisher) were added to each sample. Samples were incubated at 37°C for 30 minutes. Then, 1.2 mL of Buffer B2 was added, and samples were mixed and incubated at 50°C for 30 minutes. At this point, each G/100 tip was equilibrated with 4 mL of QBT buffer and allowed to fully empty. Then, the clarified lysate was added to each column and left overnight to allow binding to the resin and flow-through of the remainder of cell constituents. The columns were then washed twice with 7.5 mL QC buffer. The DNA was eluted in 5 mL of pre-warmed QF buffer and precipitated by adding 3.5 mL room temperature isopropanol and inverting. The DNA was then spooled on a metal inoculating loop, transferred to a 1.5 mL tube containing 200 µL Tris-EDTA (TE) buffer, and resuspended overnight at 4°C. Initial evaluation of DNA purity and quantity was performed using the Thermo Scientific NanoDrop 2000. Sufficient DNA for sequencing was extracted from two mutants each of *E. coli* K12 and the Δ*spoT* strain and three mutants each of the Δ*spoT*/Δ*relA* strain and NEB 5-alpha.

## Genome sequencing, assembly, and analysis

Illumina libraries were constructed using the Illumina Nextera XT protocol and sequenced using the Illumina NovaSeq 6000 platform to a depth of approximately 1.5 gigabases per sample. Illumina reads for parental and mutant strains were processed using Trimmomatic version 0.39 (61), then aligned against the *E. coli* K12 reference genome (NCBI Genbank accession GCA_000005845.2) using BWA Mem version 0.7.17 (62). Single nucleotide polymorphisms (SNPs) and structural variation, like insertions and deletions, were called using Pilon version 1.23 (36). SNPs identified as "Passing" by Pilon were used in downstream analyses, except when SNPs were common to parental and mutant strains. Regions with variable length indels in both the parental and descendent strains, as well as variants identified by Pilon as duplications, were also excluded.

Oxford Nanopore Technologies (ONT) long-read sequencing libraries were constructed using the Oxford Nanopore kit SQK-LSK109. Samples were barcoded and run in batches of 12 on a GridION machine (Oxford Nanopore Technologies Ltd, Science Park, UK). The initial processing of reads was performed as previously described (53). To call methylation states on the ONT data, we ran Guppy version 6.1.7 (https://community.nanoporetech.com) with the Rerio (https://github.com/nanoporetech/rerio) model res_dna_r941_min_modbases-all-context_v001.cfg, outputting 5mC and 6mA methylation information aligned to the *E. coli* K12 reference genome. We used modbam2bed

(https://github.com/epi2me-labs/modbam2bed) to further process the output from Guppy. Sites in which methylation patterns differed in AVI-resistant strains compared to parent strains were identified using the following criteria: (i) the parent strain had coverage of ≥3 reads at the position (average was ~85 reads/position), (ii) the difference in percent methylation between parent and mutant strain was at least 50%, and (iii) at least two different mutant samples met these criteria. Methylation differences in intergenic regions were considered to have potentially affected both adjacent genes.

Gene annotations were those provided in the NCBI sequence of *E. coli* K12 substr. MG1655 (NCBI Genbank identifier U00096).

## Murine thigh infection models

### Neutropenic thigh infection model

Twelve female CD-1 (ICR) mice (Charles River Laboratories, Cambridge, MA) weighing 25–30 g were treated with cyclophosphamide (European Pharmacopoeia Reference Standard) by intraperitoneal injection (150 mg/kg on day 4 and 100 mg/kg on day 1) to induce neutropenia (63, 64). On day 3, mice were treated with 5 mg/kg uranyl nitrate to cause renal impairment simulating human drug clearance (65). On day 0, a suspension of approximately $1 \times 10^7$ CFU/mL of *K. pneumoniae* FDA-CDC 0636 was prepared in sterile endotoxin-free 0.9% sodium chloride (Teknova, Hollister, CA). Mice were anesthetized with isoflurane and injected in the right thigh with 100 µL of the bacterial suspension (total $1 \times 10^6$ CFU/thigh). Four of the mice were then sacrificed by $CO_2$ inhalation for baseline colony enumeration. In one mouse, the thigh injection was inadvertently performed subcutaneously; data from this mouse were not used in subsequent calculations. Following the sacrifice, the right thigh was dissected, suspended in 1 mL sterile 0.9% sodium chloride, and emulsified in a tissue grinder. A sample of the liquid homogenate was then removed for serial 10-fold dilutions in 0.9% sodium chloride, plating, and colony enumeration using the drop method described above. Approximately 3 hours after the time of bacterial thigh infection, the remaining mice were injected subcutaneously with either 250 mg/kg of AVI dissolved in 0.9% sodium chloride (four mice) or an equivalent volume of sodium chloride alone (four mice); doses were repeated twice after this at 8-hour intervals for a total of three doses. AVI dosing was selected based on the highest AVI dosing reported in the literature in mouse thigh models of AVI in combination with ceftazidime (66). At 24 hours after the first dose, mice were euthanized, and thighs were removed for colony enumeration as described above. One mouse in the AVI treatment group developed lethargy and apparent seizure activity shortly after the third AVI injection, at which time it was sacrificed; thigh dissection and plating were performed immediately following sacrifice.

### Non-neutropenic thigh infection model

To determine the bacterial burden required to establish a thigh infection with *K. pneumoniae* FDA-CDC 0636 in a non-neutropenic mouse, two mice each were injected with three different bacterial inocula ($1 \times 10^6$, $1 \times 10^7$, and $1 \times 10^8$ CFU/thigh) as described above. After 24 hours, mice were euthanized, and thighs were removed for colony enumeration as described above; only the highest inoculum resulted in an increase in CFU/thigh over the 24-hour period (increase in CFU/thigh of 1.3 $\log_{10}$ vs decrease of ~1.8 $\log_{10}$ CFU/thigh with the two lower inocula). The treatment experiment was performed as for the neutropenic thigh infection model, except that pre-treatment with cyclophosphamide was omitted and the $10^8$ CFU/thigh inoculum was used for infection. There were five mice each in the baseline, AVI treatment, and saline treatment groups.

All mouse experiments were performed under an institutional animal care and use committee-approved protocol.

## Data analysis

Statistical analysis was performed using GraphPad Prism 10 software. Growth curve data analysis was performed using GrowthRates 6.2.1 (Bellingham Research Institute) (60).

## ACKNOWLEDGMENTS

We would like to thank Terrence Shea for helpful discussions. This project has been funded in part with federal funds from the National Institute of Allergy and Infectious Diseases, National Institutes of Health, Department of Health and Human Services, under grant numbers K08AI132716 and R01AI178875 to Thea Brennan-Krohn, and U19AI110818 to the Broad Institute. Some assays in this work were performed on equipment provided by the Massachusetts Life Science Novel Therapeutics Delivery Grant to James Kirby, Beth Israel Deaconess Medical Center.

## AUTHOR AFFILIATIONS

[1]Department of Pathology, Beth Israel Deaconess Medical Center, Boston, Massachusetts, USA

[2]Infectious Disease and Microbiome Program, Broad Institute of MIT and Harvard, Cambridge, Massachusetts, USA

[3]Division of Infectious Diseases, Boston Children's Hospital, Boston, Massachusetts, USA

[4]Harvard Medical School, Boston, Massachusetts, USA

## PRESENT ADDRESS

Michelle Nägeli, Klinik Favoriten, Vienna, Austria

Shade Rodriguez, Pathobiology Graduate Program, Brown University, Providence, Rhode Island, USA

## AUTHOR ORCIDs

Abigail L. Manson ⓘ http://orcid.org/0000-0002-3800-0714
Ashlee M. Earl ⓘ http://orcid.org/0000-0001-7857-9145
Thea Brennan-Krohn ⓘ http://orcid.org/0000-0002-1284-0643

## FUNDING

| Funder | Grant(s) | Author(s) |
| --- | --- | --- |
| National Institute of Allergy and Infectious Diseases | K08AI132716 | Thea Brennan-Krohn |
| National Institute of Allergy and Infectious Diseases | R01AI178875 | Thea Brennan-Krohn |
| National Institute of Allergy and Infectious Diseases | U19AI110818 | Abigail L. Manson |
| | | Ashlee M. Earl |

## AUTHOR CONTRIBUTIONS

Michelle Nägeli, Data curation, Investigation, Methodology, Writing – original draft | Shade Rodriguez, Data curation, Investigation, Methodology, Writing – original draft | Aimee Iradukunda, Formal analysis, Investigation, Writing – review and editing | Abigail L. Manson, Conceptualization, Data curation, Formal analysis, Methodology, Writing – original draft, Writing – review and editing | Ashlee M. Earl, Conceptualization, Data curation, Formal analysis, Methodology, Writing – review and editing | Thea Brennan-Krohn, Conceptualization, Data curation, Formal analysis, Funding acquisition, Investigation, Methodology, Project administration, Supervision, Writing – original draft, Writing – review and editing

## DATA AVAILABILITY

Both Illumina and Oxford Nanopore sequencing reads were deposited at the Sequence Read Archive under Bioproject PRJNA1140646.

## ADDITIONAL FILES

The following material is available online.

### Supplemental Material

**Tables S1 to S4 (Spectrum03241-24-s0001.xlsx).** Supplemental tables.

### Open Peer Review

**PEER REVIEW HISTORY (review-history.pdf).** An accounting of the reviewer comments and feedback.

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
