## [Reviewer comments · Microbiology Spectrum]

Microbiology Spectrum

Rapid Emergence of Resistance to Broad-Spectrum Direct Antimicrobial Activity of Avibactam

Michelle Nägeli, Shade Rodriguez, Aimee Iradukunda, Abigail Manson, Ashlee Earl, and Thea Brennan-Krohn

Corresponding Author(s): Thea Brennan-Krohn, Beth Israel Deaconess Medical Center

Review Timeline:

Submission Date:	January 10, 2025
Editorial Decision:	February 24, 2025
Revision Received:	April 16, 2025
Accepted:	April 28, 2025

Editor: Antonio Ruzzini

Reviewer(s): The reviewers have opted to remain anonymous.

Transaction Report:

DOI: <https://doi.org/10.1128/spectrum.03241-24>

Re: Spectrum03241-24 (Rapid Emergence of Resistance to Broad-Spectrum Direct Antimicrobial Activity of Avibactam)

Dear Dr. Thea Brennan-Krohn:

Thank you for the privilege of reviewing your work. Below you will find my comments, instructions from the Spectrum editorial office, and the reviewer comments.

As part of the transfer to Microbiology Spectrum, the previous reviewer comments were considered as were the changes between submissions. The transferred work was subject of a second review, which I viewed as positive; however, there are some elements that require clarification and language that should be tempered to be better reflect the data/current state of knowledge.

Revision Guidelines

Sincerely,
Antonio Ruzzini
Editor
Microbiology Spectrum

Reviewer #1 (Comments for the Author):

Overall, the work is of interest and concerningly, suggests that prior exposure to avibactam may lead to resistance to drugs in

development that target PBP2 (e.g., nacubactam, zidebactam) and others that are already available (mecillinam, durlobactam). One question that is not addressed in this manuscript, and is of significant importance, is whether the emergence of resistance seen in this manuscript is abrogated by the addition of ceftazidime - to my knowledge, PBP-2 specific mechanisms of resistance have not been described as occurring in response to the combination of ceftazidime-avibactam, where resistance is largely due to alterations in beta lactamase structure or expression levels. It would be useful to determine if the avibactam-resistant isolates here have any altered response to the ceftazidime-avibactam combination and if continued exposure to ceftazidime-avibactam in ceftazidime-avibactam resistant isolates leads to the mutations observed in this study.

Line 278 (revised manuscript): "Avibactam appears to have greater in vivo efficacy in an immunocompetent mouse model" The section header should make clear what the comparator is, as the comparator in this sentence could either be another drug or another model system. The overall finding that antimicrobials perform better in immunocompetent mice is not novel however, and it is not clear how this adds to the manuscript in any meaning way.

Reviewer #2 (Comments for the Author):

The manuscript by Nägeli et al. examines avibactam resistance in Enterobacterales. While the study investigates avibactam-resistant mutants using whole genome sequencing, DNA methylation and MIC determination, several concerns reduce its significance and clinical relevance.

The use of non-standard methodologies that do not align with CLSI guidelines raises questions about reproducibility. Furthermore, the choice of DBOs is suboptimal; the inhibition profile of avibactam is not completely aligned with the drugs that the authors are discussing and neither zidebactam nor nacubactam are clinically approved. While the study examines mutations across diverse strains, the conclusions about PBP2-targeting drugs may not be fully supported by the data.

Line 6, line 129 and elsewhere: "...mediated by inhibition of penicillin-binding protein 2 5 (PBP2). This activity is mechanistically similar to that of more potent novel DBOs (zidebactam, 6 nacubactam) in late clinical development..." There are significant variations in the concentrations required for half-maximal inhibition of PBP2, and the inhibition profiles differ as well. zidebactam specifically targets PBP2, whereas avibactam exhibits a greater affinity for PBP1b and PBP4.

Line 151: The molecular mechanisms have not been detailed. It is crucial to characterize the beta-lactamases present in strains that exhibit resistance to Zidebactam, as its activity spectrum does not encompass the majority of β -lactamases encoded by mobile genetic elements.

Line 368: "...rendering PBP2 apparently inessential...". This represents a significant assumption that lacks experimental validation.

Line 276 and elsewhere: AVI-resistance does not have established breakpoints for avibactam when used independently; therefore, a more accurate term would be diminished sensitivity.

Avibactam is not used as a standalone medication for patients. Therefore, the investigation of resistance to these combinations must consider the contributions of the β -lactam drug as well as the β -lactamase inhibitor properties of the DBO, alongside the direct activity of the DBO itself.

Line 151: The molecular mechanisms have not been detailed. It is crucial to characterize the beta-lactamases present in strains that exhibit resistance to Zidebactam, as its activity spectrum does not encompass the majority of β -lactamases encoded by mobile genetic elements.

Line 368: "...rendering PBP2 apparently inessential...". This represents a significant assumption that lacks experimental validation.

Line 276 and elsewhere: AVI-resistance does not have established breakpoints for avibactam when used independently; therefore, a more accurate term would be diminished sensitivity.

Avibactam is not used as a standalone medication for patients. Therefore, the investigation of resistance to these combinations must consider the contributions of the β -lactam drug as well as the β -lactamase inhibitor properties of the DBO, alongside the direct activity of the DBO itself.

We appreciate the reviewers' thoughtful feedback, and have responded to their suggestions and questions in the manuscript as outlined below. An additional author, Aimee Iradukunda, has also been added to the manuscript; Dr. Iradukunda performed and analyzed additional testing suggested by Reviewer #1 and reviewed the current manuscript.

Reviewer #1 (Comments for the Author):

Overall, the work is of interest and concerningly, suggests that prior exposure to avibactam may lead to resistance to drugs in development that target PBP2 (e.g., nacubactam, zidebactam) and others that are already available (mecillinam, durlobactam). One question that is not addressed in this manuscript, and is of significant importance, is whether the emergence of resistance seen in this manuscript is abrogated by the addition of ceftazidime - to my knowledge, PBP-2 specific mechanisms of resistance have not been described as occurring in response to the combination of ceftazidime-avibactam, where resistance is largely due to alterations in beta lactamase structure or expression levels. It would be useful to determine if the avibactam-resistant isolates here have any altered response to the ceftazidime-avibactam combination and if continued exposure to ceftazidime-avibactam in ceftazidime-avibactam resistant isolates leads to the mutations observed in this study.

Author response: We appreciate the reviewer's thoughtful comments, and we agree that the role of the beta-lactam will be extremely important in understanding how resistance to novel beta-lactam/DBO combinations emerges. In response to the reviewer's suggestions, we tested the activity of ceftazidime and ceftazidime-avibactam against avibactam-resistant isolates (lines 146-148 of marked-up manuscript; lines 145-147 of clean manuscript.) A more complete exploration of the role of beta-lactam pairings is outside the scope of the present work, in which we were focused on investigating direct activity of avibactam alone, and is likely to be more relevant for newer, more potent DBOs that exert direct activity at clinically administered concentrations. We are currently undertaking such investigations in our lab and the results will be the subject of future manuscripts.

Line 278 (revised manuscript): "Avibactam appears to have greater *in vivo* efficacy in an immunocompetent mouse model" The section header should make clear what the comparator is, as the comparator in this sentence could either be another drug or another model system. The overall finding that antimicrobials perform better in immunocompetent mice is not novel however, and it is not clear how this adds to the manuscript in any meaningful way.

Author response: The section header has been changed to read as follows: "Avibactam appears to have greater *in vivo* efficacy in an immunocompetent mouse model compared to a neutropenic model." (Lines 265-266 of marked-up manuscript; lines 263-264 of clean manuscript.) We recognize that the finding of better performance of avibactam in an immunocompetent model relative to an immunocompromised model is not novel, but we felt that this result suggested the specific possibility of impaired fitness of avibactam-resistant mutants in an *in vivo* setting. Potentially supporting this hypothesis, we tested the avibactam MICs of bacteria recovered from mice in the immunocompetent model and found them unchanged. We had not included this data in

the original manuscript but have added it here (lines 273-275 of marked-up manuscript; lines 271-273 of clean manuscript) to clarify our rationale for these experiments, and have also added a phrase in the discussion to discuss this (lines 389-390 of marked-up manuscript; lines 387-388 of clean manuscript.)

Reviewer #2 (Comments for the Author):

The manuscript by Nägeli et al. examines avibactam resistance in Enterobacterales. While the study investigates avibactam-resistant mutants using whole genome sequencing, DNA methylation and MIC determination, several concerns reduce its significance and clinical relevance.

The use of non-standard methodologies that do not align with CLSI guidelines raises questions about reproducibility. Furthermore, the choice of DBOs is suboptimal; the inhibition profile of avibactam is not completely aligned with the drugs that the authors are discussing and neither zidebactam nor nacubactam are clinically approved. While the study examines mutations across diverse strains, the conclusions about PBP2-targeting drugs may not be fully supported by the data.

Author response:

- With regard to CLSI guidelines, the only technique used in our work to which CLSI methods are applicable is minimal inhibitory concentration (MIC) testing. While the assay we used (the digital dispensing method) is not a CLSI-approved method, it is an adaptation of gold-standard broth microdilution testing, and previous work in our laboratory has shown it to have very high precision as well as accuracy compared to broth microdilution (Smith KP, Kirby JE. 2016. Verification of an Automated, Digital Dispensing Platform for At-Will Broth Microdilution-Based Antimicrobial Susceptibility Testing. *J Clin Microbiol* 54:2288–2293; PMID 27335151). A similar version of our method, using 96-well instead of 384-well plates, was adopted by the CDC to perform testing of isolates for aztreonam-avibactam for clinical use (<https://arpsp.cdc.gov/profile/exast-arln/2019>; Ransom E et al. 2020. Validation of Aztreonam-Avibactam Susceptibility Testing Using Digitally Dispensed Custom Panels. *J Clin Microbiol* 58:e01944-19; PMID 32051259).
- We appreciate that the findings in this work may not be identical to those discovered when other, newer DBOs are tested. We have adjusted the language in the manuscript to emphasize this point further, and to argue that our findings support the continued investigation of these new compounds rather than providing any conclusions about them, as follows:
 - “This activity is mechanistically similar to that of more potent novel DBOs (zidebactam, nacubactam) in late clinical development” changed to, “This activity has some mechanistic similarities to that of more potent novel DBOs (zidebactam, nacubactam) in late clinical development.” (Lines 6-7 of marked-up and clean manuscript.)

- “Our results emphasize the importance of further study of resistance to novel DBO agents as they approach clinical availability” changed to, “Our results emphasize the importance of further study of resistance to novel DBO agents as they approach clinical availability in order to determine whether similar patterns of resistance may be observed with these compounds.” (Lines 95-97 of marked-up manuscript; lines 94-96 of clean manuscript.)

Line 6, line 129 and elsewhere: "...mediated by inhibition of penicillin-binding protein 2 5 (PBP2). This activity is mechanistically similar to that of more potent novel DBOs (zidebactam, 6 nacubactam) in late clinical development..." There are significant variations in the concentrations required for half-maximal inhibition of PBP2, and the inhibition profiles differ as well. zidebactam specifically targets PBP2, whereas avibactam exhibits a greater affinity for PBP1b and PBP4.

Author response: As noted above, we recognize that the findings in this work will likely not be identical to those discovered when newer DBOs are evaluated by similar methods, and we have adjusted our wording as noted above. However, we feel that, given the similarities that do exist between avibactam and newer DBOs, our findings suggest that research into mutation frequencies and resistance mechanisms for these new drugs will be important to study. (I looked for articles describing PBP1b-binding of avibactam, but all the assays I can see reported in published work describe undetectable binding of PBP1b by avibactam. One study did describe weak binding of PBP4, a low-molecular mass PBP that carries out a different role in the cell than PBPs 1-3, such that it seems unlikely that this function is a major contributor to the primarily PBP2-dependent activity of avibactam).

Line 151: The molecular mechanisms have not been detailed. It is crucial to characterize the beta-lactamases present in strains that exhibit resistance to Zidebactam, as its activity spectrum does not encompass the majority of β -lactamases encoded by mobile genetic elements.

Author response: The line the reviewer is referring to is at the end of the following sentence: “Notably, bacterial cells that are resistant to PBP2 targeting drugs such as mecillinam and nacubactam exhibit rounding during treatment with the drug, even though they are still able to survive and replicate (15, 25, 26), as resistance typically involves compensatory mutations in genes other than PBP2.” I believe the reviewer is referring to resistance to mecillinam and nacubactam, as zidebactam is not discussed here. We have clarified that strains in the papers referenced did not produce beta-lactamases. (Lines 125-129 of marked-up manuscript and lines 124-128 of clean manuscript)

Line 368: "...rendering PBP2 apparently inessential...". This represents a significant assumption that lacks experimental validation.

Author response: The referenced article (Thulin E, Andersson DI. 2019. Upregulation of PBP1B and LpoB in cysB Mutants Confers Mecillinam (Amdinocillin) Resistance in Escherichia coli. AAC. PMID 31332059) involved experimental study of the mechanism described, although we have qualified the result as “apparently” inessential to reflect

the fact that further investigation may be needed to definitely establish the certainty of the finding.

Line 276 and elsewhere: AVI-resistance does not have established breakpoints for avibactam when used independently; therefore, a more accurate term would be diminished sensitivity. Avibactam is not used as a standalone medication for patients. Therefore, the investigation of resistance to these combinations must consider the contributions of the β -lactam drug as well as the β -lactamase inhibitor properties of the DBO, alongside the direct activity of the DBO itself.

Author response: We appreciate the reviewer's point that there are no clinically established breakpoints for avibactam. We have explained this and defined our use of the term "resistant" in the text, and we would argue that using this term rather than "diminished sensitivity" increases readability of the text, particularly as the term is used repeatedly throughout the manuscript. (Lines 83-86 of marked-up manuscript; lines 82-85 of clean manuscript: "Because no clinical interpretive criteria for direct avibactam activity exist, we use the term "AVI resistance" throughout to refer to strains that developed AVI MICs of $\geq 8x$ the starting MIC upon exposure to AVI, while the parent strains are referred to as "AVI susceptible".")

Re: Spectrum03241-24R1 (Rapid Emergence of Resistance to Broad-Spectrum Direct Antimicrobial Activity of Avibactam)

Dear Dr. Thea Brennan-Krohn:

Thank you for providing a revised manuscript addressing reviewer comments and suggestions, including the addition a new experimental results.

I am pleased to inform you that your manuscript has been accepted, and I am forwarding it to the ASM production staff for publication. Your paper will first be checked to make sure all elements meet the technical requirements. ASM staff will contact you if anything needs to be revised before copyediting and production can begin. Otherwise, you will be notified when your proofs are ready to be viewed.

Sincerely,
Antonio Ruzzini
Editor
Microbiology Spectrum